# Seroprevalence of immunoglobulin G antibodies against SARS-CoV-2 in Cyprus

Christos Papaneophytou[1], Andria Nicolaou[2], Myrtani Pieri[1], Vicky Nicolaidou[1], Eleftheria Galatou[1], Yiannis Sarigiannis[1], Markella Pantelidou[2], Pavlos Panayi[2], Theklios Thoma[2], Antonia Stavraki[2], Xenia Argyrou[2], Tasos Kalogiannis[2], Kyriacos Yiannoukas[2], Christos C. Petrou[1], Kyriacos Felekkis[1] *

1 Department of Life and Health Science, School of Sciences and Engineering, University of Nicosia, Nicosia, Cyprus, 2 Yiannoukas Medical Laboratories/ Bioiatriki Group, Nicosia, Cyprus

* felekkis.k@unic.ac.cy

## Abstract

Monitoring the levels of IgG antibodies against the SARS-CoV-2 is important during the coronavirus disease 2019 (COVID-19) pandemic, to plan an adequate and evidence-based public health response. After this study we report that the plasma levels of IgG antibodies against SARS-CoV-2 spike protein were higher in individuals with evidence of prior infection who received at least one dose of either an mRNA-based vaccine (Comirnaty BNT162b2/ Pfizer-BioNTech or Spikevax mRNA-1273/Moderna) or an adenoviral-based vaccine (Vaxzervia ChAdOx1 nCoV-19 /Oxford-Astra Zeneca) (n = 39) compared to i) unvaccinated individuals with evidence of prior infection with SARS-CoV-2 (n = 109) and ii) individuals without evidence of prior infection with SARS-CoV-2 who received one or two doses of one of the aforementioned vaccines (n = 342). Our analysis also revealed that regardless of the vaccine technology (mRNA-based and adenoviral vector-based) two doses achieved high anti-SARS-CoV-2 IgG responses. Our results indicate that vaccine-induced responses lead to higher levels of IgG antibodies compared to those produced following infection with the virus. Additionally, in agreement with previous studies, our results suggest that among individuals previously infected with SARS-CoV-2, even a single dose of a vaccine is adequate to elicit high levels of antibody response.

## Introduction

The Severe Acute Respiratory Syndrome Coronavirus 2 (SARS-CoV-2), which causes COVID-19 continues to spread worldwide as a severe ongoing pandemic. Immunity to SARS-CoV-2, induced either through natural infection or vaccination, has been demonstrated to afford a degree of protection against reinfection and transmission of the virus and/or reduce the risk of clinically significant outcomes [1]. Antibodies against SARS-CoV-2 are an essential part of immunity against the virus, as an appropriate neutralizing response can efficiently block virions from successfully infecting Angiotensin-converting enzyme 2 (ACE-2) receptor-expressing cells. Understanding the humoral immune response and analyzing the antibody profiles induced against SARS-CoV-2 can guide public health measures and control strategies.

**Data Availability Statement:** The data underlying the results presented in the study are available from (https://doi.org/10.5281/zenodo.5992410).

**Funding:** The study was supported internally by the University of Nicosia and Yiannoukas Medical Laboratories/ Bioiatriki Group. The funders had no role in study design, data collection and analysis, decision to publish, or preparation of the manuscript. The authors did not receive any salary from the funders as part of this work.

**Competing interests:** The authors have declared that no competing interests exist.

However, antibody response against the SARS-CoV-2 is still a subject of debate and must be addressed carefully [2]. The protective role of antibodies against the virus and its variants remains unknown. However, such antibodies usually demonstrate a sufficient correlation of antiviral immunity, and anti–receptor-binding domain antibody levels correspond to plasma viral neutralizing activity [3].

The SARS-CoV-2 infection produces early detectable humoral immune responses in most patients, leading to the development of neutralizing antibodies (NAbs) in the vast majority of cases [4–7]. However, the duration, magnitude, and protective capacity of the humoral immune response remain elusive to date [8]. Seroprevalence has been reported to be low in the general population and varied among countries and territories. In addition, recent studies have reported a decline in neutralization titer with time for up to 8 months after SARS-CoV-2 infection [9–11]. Other studies have shown the induction of neutralizing and protective anti-SARS-CoV-2 antibodies after infection, which reduced the risk of reinfection for the following 13 months [12]. However, the long-term time course of the antibody response in COVID-19 disease is not yet fully determined. Some studies show a significant decrease in antibody concentrations within 3–4 months from the onset of symptoms [3, 13, 14]. Other reports find constant or only slightly decreased levels, starting from 4 months and up to 10 months from symptom onset [4–7], even when specific neutralizing antibodies [8] were measured. In particular, the time course of the antibody response seems variable, also according to the method used [7, 9]. On the other hand, antibodies seem to persist through 4–6 months in vaccinated individuals [10, 11].

Several SARS-CoV-2 vaccines have been developed and the European Medicines Agency (EMA), has utilised the rolling review regulatory tool to speed up the assessment process during a public health emergency. To date, EMA has granted five conditional approvals for vaccines that met a positive benefit-risk balance: BNT162b2 (Comirnaty, Pfizer/BioNTech), mRNA-1273 (Spikevax Moderna), ChAdOx1-S (Vaxzevria, AstraZeneca and the University of Oxford Vaxzevria), Ad26.COV2-S (Jahnsen—Johnson and Johnson), [15] and NVX-CoV2373 (Nuvaxovid, Novavax) [16]. These vaccines offer protection against SARS-CoV-2 by generating immune responses against the spike antigen of the virus. An ideal SARS-CoV-2 vaccine should prevent infection and protect from severe disease in all vaccinated populations, as well as elicit long-term memory immune responses after a minimal number of immunizations or booster doses [17]. On the other hand, regulatory agencies recommend that the primary endpoint should be how well a vaccine prevents laboratory-confirmed COVID-19 disease (symptomatic disease) of any severity and not whether a vaccine provides sterilizing immunity [18]. Although vaccines elicit a strong and protective immune response [19], the potency of such response relative to the response induced by infection is not well understood [20]. Moreover, the potency and durability of infection-induced SARS-CoV-2 immunity have crucial implications for reinfection and vaccine effectiveness [8].

A major question is whether vaccine-induced responses may be more durable, in terms of long-term protection, than those following infection [1]. Assessment of the kinetics of SARS-CoV-2 antibodies is essential in predicting protection against reinfection and durability of vaccine protection [12]. Naturally, antibody profiles and dynamics vary depending on the Ig isotype of interest [21]. However, evidence has also been gathered indicating that there are dissimilar profiles of antibody kinetics depending on the antigen of interest, and consequently, natural antibody levels vary significantly and cannot be characterized as a whole [2]. It should be noted that the humoral response (i.e., antibodies) is not the only protective response against SARS-CoV-2 and other infections, and that cell-mediated immunity (i.e. T cells) may be maintained despite putative lack of detection in serum antibody levels [22].

Although it is still not clear whether antibodies against SARS-CoV-2 correlate with protective immunity, a study of SARS-CoV-2 antibodies among residents of certain geographical regions can aid in the estimation of the immunological protection against subsequent infection (see [11] and references cited therein). Furthermore, knowledge of the magnitude, timing, and longevity of antibody responses after SARS-CoV-2 infection is vital for understanding the role that antibodies might play in disease clearance and protection from reinfection and/or severe disease. Finally, as huge emphasis has been placed on antibody reactivity assays to determine seroprevalence against SARS-CoV-2 in the community to estimate infection rates, it is vital to understand immune responses after infection to define parameters in which antibody tests can provide meaningful data in the absence of PCR testing in population studies [23].

Even though the epidemic of COVID-19 in Cyprus started in Feb 2020, no seroprevalence data are available thus far. Furthermore, on 28th December 2020, Cyprus began its national vaccination program with the Comirnaty vaccine followed by the approval of Spikevax and Vaxzervia vaccines. The current study has assessed the levels of SARS-CoV-2 antibodies in the general population of Cyprus. Specifically, we compared antibody levels among three groups of participants: i) vaccinated without evidence of previous infection, ii) unvaccinated with evidence of previous infection, and iii) vaccinated with evidence of previous infection. Blood samples were analyzed by enzyme-linked immunosorbent assay (ELISA) to detect anti–SARS-CoV-2 spike receptor-binding domain IgG antibodies.

## Methods

### Population

In this nationwide study, we enrolled 702 individuals from four cities (Nicosia, Limassol, Larnaca, and Paphos) in Cyprus from May to November 2021. Demographic data, including age, gender, place of residence, and occupation of each participant, were collected during recruitment. The study has been approved by the Cyprus National Bioethics Committee (EEBK/EΠ/2021/06). All individuals provided written consent to participate. In detail, blood samples were obtained aseptically from each participant at Yiannoukas Medical Laboratories/Bioiatriki Group throughout different locations in Cyprus. Participants visited Yiannoukas Medical Laboratories/Bioiatriki Group to provide blood for analysis for a routine check-up or other tests prescribed by their physicians. These individuals were informed about the study and its aims and asked if they were willing to donate an additional 5–9 mL of blood specifically for this study. Individuals that agreed were given the consent form to sign and also completed a short questionnaire requesting demographic data as well as whether the participant had ever tested positive for COVID-19 with either an antigen rapid test and/or an RT-qPCR test for SARS-CoV-2 or had received vaccination against COVID-19. Furthermore, only participants with a negative antigen rapid test and/or an RT-qPCR test for SARS-CoV-2 at the time of blood collection were eligible to enroll in this study.

### Laboratory testing

**Blood collection, handling, and storage.** Five (5) to nine (9) mL of whole blood were collected from volunteers using needles of diameter >23 gauge to prevent hemolysis and were immediately transferred into commercially available plasma (clot activator) tubes. Each tube was labeled with a unique code. Tubes were inverted carefully 10 times to mix blood and anticoagulant and stored at 4°C until centrifugation according to the rules proposed by the Standard Operating Procedures Internal Working Group (SOPIWG)/ Early Detection Research Network (EDRN) for specimen collection (including blood samples) [24]. To separate the plasma from cells, samples were centrifuged at 1500g for 20 min. Following centrifugation, the

plasma was transferred into clean tubes using a sterile serological pipette. Samples were then maintained at 2–8˚C until further processing.

**Antibodies measurement.** Specific anti-SARS-CoV-2-IgG levels were determined by the SIEMENS Dimension EXL system which employs a chemiluminescent immunoassay based on Luminescent oxygen channeling assay (LOCI) technology. The LOCI reagents include two synthetic bead reagents (Sensibeads and Chemibeads) and a biotinylated anti-human IgG antibody. Sensibeads are coated with streptavidin and contain photosensitizer. Chemibeads are coated with an anti-- Fluorescein isothiocyanate (FITC) antibody and contain chemiluminescent dye. Furthermore, the anti-FITC antibody-coated-Chemibeads are pre-decorated with fluoresceinated S1 receptor-binding domain (RBD) antigen of the spike protein of the SARS-CoV-2 virus. All components for the detection of anti-IgG against SARS-CoV-2 including the fluoresceinated S1 receptor-binding RBD antigen were included in the Dimension Vista SARS CoV 2 IgG (COV2G) assay (Siemens Healthcare Diagnostics Inc., Erlangen, Germany, Cat #K7771

11417771). The sample is incubated with Chemibeads for 1 minute and subsequently, the biotinylated antibody is added to form bead-CoV-2 antigen-biotinylated antibody sandwiches. After the completion of incubation, Sensibeads are added to bind to the biotin to form bead-pair immunocomplexes. Illumination of the complex at 680nm generates singlet oxygen from Sensibeads which diffuses into the Chemibeads, triggering a chemiluminescent reaction. The resulting signal was measured at 612nm and was proportional to IgG concertation in the sample. IgG levels were determined by the semiquantitative mode of the SIEMENS Dimension EXL system using a 5-level LOGIT calibration curve and the results were presented as relative (Ind) Units. The Dimension EXL system cutoff analyte value was 1000 relative Units and it was used to identify IgG positive samples. The sensitivity of the method (for samples obtained ≥ 14 days post-infection with SARS-CoV-2) was 100% (95% CI, 95.9–100%) and the specificity was 100% (95% CI, 95.8–100%).

## Statistical analysis

Continuous variables with normal distribution were expressed as means ± standard deviation (SD) and categorical variables as counts and percentages). The normal distribution of continuous data was analyzed with the D'Agostino & Pearson omnibus normality test. Antibody levels were log-transformed before all statistical processes. IgG levels were not normally distributed, and thus are presented as median and interquartile ranges (IQR). We used the Mann-Whitney $U$- test and the Pearson′s Chi-square test to compare the IgG levels between groups. The linear relationship of IgG responses with the age of participants was carried out using Spearman's correlation coefficient. Correlations were classified as very weak (correlation coefficient ($r$) < 0.20), weak ($r = 0.20$–$0.39$), moderate ($r = 0.40$–$0.59$), strong ($r = 0.60$–$0.79$), or very strong ($r > 0.80$). Statistical significance was set at $p < 0.05$. Statistical analysis was performed using GraphPad Prism (v.8.2, GraphPad Software Inc., San Diego, CA, USA).

## Results

### Demographics and vaccination coverage

A total of 702 individuals participated in this study. Descriptive data for the study population are summarized in Table 1. The median (IQR) age was 43 years (33–59), 332 participants (47.3%) were men and 370 (52.7%) participants were women. Overall, 148 (21.1%) participants with a SARS-CoV-2 positive PCR-test or a positive nose/throat swab rapid test were considered to have previously been infected with SARS-CoV-2 regardless of whether they had reported previous symptoms, while 39 out of the 148 had received one or two doses of a

**Table 1. Characteristics of the 702 participants.**

| Variable | All participants (n = 702) | No prior infection | | Prior infection | | |
|---|---|---|---|---|---|---|
| | | Unvaccinated | Vaccinated | Unvaccinated | Vaccinated | p-value[1] |
| | | (n = 212) | (n = 342) | (109) | (n = 39) | |
| Age, median (IQR) | 43 (33–59) | 38 (30–51) | 49 (36–63) | 40 (31–55) | 45 (33–58) | <0.0001 |
| Age Range | | | | | | |
| 18–45, n (%) | 385 (54.8) | 140 (66.0) | 158 (46.2) | 66 (60.6) | 21 (53.8) | <0.0001 |
| 46–65, n (%) | 222 (31.7) | 58 (27.4) | 114 (33.3) | 35 (32.1) | 15 (38.5) | |
| >66, n (%) | 95 (13.5) | 14 (6.6) | 70 (20.5) | 8 (7.3) | 3 (7.7) | |
| Gender | | | | | | |
| Male, n (%) | 332 (47.3) | 102 (48.1) | 160 (46.8) | 48 (44.0) | 22 (56.4) | >0.05 |
| Female, n (%) | 370 (52.7) | 110 (51.9) | 182 (53.2) | 61 (56.0) | 17 (43.6) | |

Data are presented as median (IQR) or n (%)

[1] Difference among all types. Differences in measurement data among the four groups were compared with the Kruskal Wallis test followed by Dunn's multiple comparisons test or Chi-squared test

licensed vaccine. From a total of 554 participants without evidence of previous infection, 339 had received either one or two doses of a licensed vaccine, 2 participants had received 3 doses of Pfizer/BioNTech and 1 participant had received 3 doses of the Moderna vaccine. Based on these definitions, we initially categorized participants into three groups as follows: i) vaccinated without evidence of prior infection (n = 342); ii) unvaccinated with evidence of prior infection (n = 109), and iii) vaccinated with evidence of prior infection (n = 39). Unvaccinated participants without evidence of prior infection (n = 212) were used as the control group. We also categorized participants into three age groups namely the 18–45 years of age (n = 385), 46–65 years of age (n = 222), and > 66 years of age (n = 95).

Table 2 summarizes the vaccination coverage in terms of vaccine type and the number of doses in individuals with or without evidence of prior infection with SARS-CoV-2. It is shown that more than 63.5% of the participants (242 of 381) received one, two, or three doses of the Pfizer/BioNTech vaccine.

Antibodies against SARS-CoV-2 increased following vaccination or infection as illustrated in Fig 1, while the levels of specific anti-spike IgG were significantly higher ($p<0.0001$) in all three groups compared to the control group (Fig 1). Specifically, without evidence of prior infection, anti-spike IgG antibodies were detected after vaccination in 312 of 342 (91.2%) participants. In the unvaccinated group with evidence of prior infection, antibodies were detected in 84 of 109 (77.1%) participants. Interestingly, in the vaccinated group with evidence of prior infection anti-spike IgG antibodies were detected in 100% of participants. Furthermore, this group (vaccinated with evidence of prior infection) also had the highest virus-specific IgG antibody levels detected as illustrated in Fig 1. Interestingly, the levels of specific anti-spike IgG

**Table 2. Vaccination coverage in terms of vaccine type and number of doses given.**

| Vaccine | No prior infection (n = 342) | | | Prior infection (n = 39) | |
|---|---|---|---|---|---|
| | One dose (n = 78) | Two doses (n = 261) | Three doses (n = 3) | One dose (n = 25) | Two doses (n = 14) |
| Oxford Astra Zeneca | 33 | 32 | 0 | 9 | 4 |
| Pfizer-BioNTech | 25 | 191 | 2 | 6 | 18 |
| Moderna | 10 | 38 | 1 | 0 | 2 |
| Johnson & Johnson | 10 | - | - | - | - |

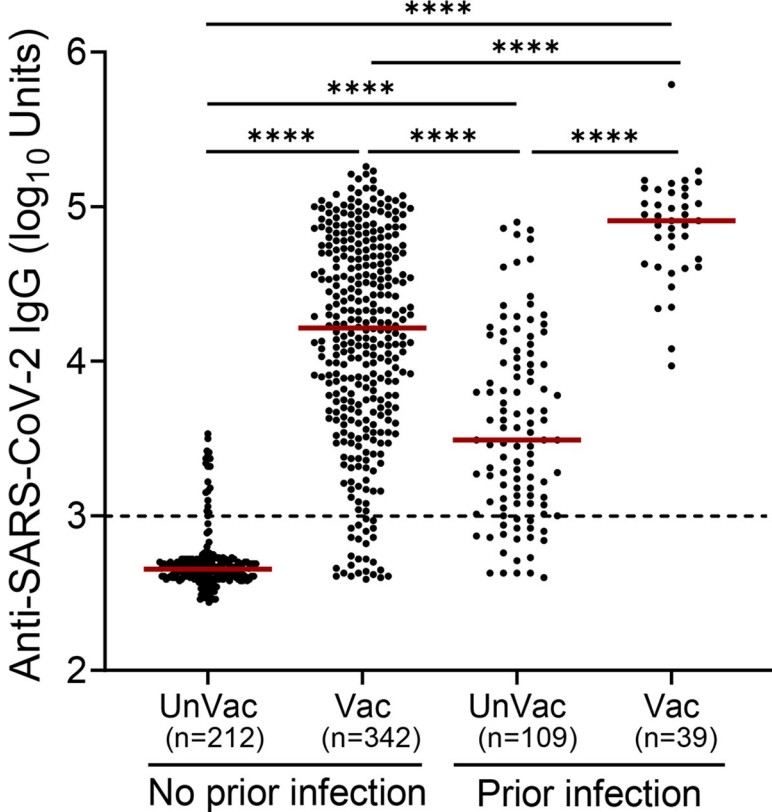

**Fig 1. Anti-spike IgG antibody responses.** Anti-SARS-CoV-2 IgG levels were determined in individuals without or with evidence of prior infection. Participants were not vaccinated (UnVac) or received at least one dose of a vaccine against SARS-CoV-2 (Vac). The number of participants (n) in each group is shown in parentheses. Statistical significance was determined using the Kruskal Wallis test followed by Dunn's multiple comparisons test. Horizontal bars indicate median values. Statistically significant differences are indicated with asterisks: $****p < 0.0001$. Dotted line: positive cut-off value.

were significantly higher ($p < 0.0001$) in the vaccinated group without evidence of previous infection (n = 342) compared to the unvaccinated group with evidence of previous infection (n = 109). There was no statistical difference in anti-spike IgG antibodies levels between men and women in all groups as resulted by the Mann Whitney $U$ test (S1 Fig). Due to the small number, individuals who received the Johnson and Johnson vaccine or 3 doses of either the Pfizer/BioNTech or the Moderna Vaccine (Table 2) were excluded from the subsequent analysis.

We then analyzed the antibody levels in participants without evidence of prior infection who received one dose or two doses of a vaccine in the three age groups (Fig 2A). After vaccination, anti-spike IgG antibodies were detected in 137 of 151 (90.7%) participants of the 18–45 years of age group. In the 46–65 years of age and >66 years of age groups, antibodies were detected in 104 of 110 (94.5%) and 65 of 68 (95.6%), respectively. Our analysis revealed that there were no statistically significant differences in specific anti-spike IgG levels among the three age groups (Fig 2A).

We also compared the levels of vaccine-elicited IgG after the administration of one and two doses of the same vaccine (Fig 2B). There was no statistically significant difference in anti-spike IgG levels between one dose and two doses of the Oxford-Astra Zeneca. Anti-spike IgG levels were significantly increased ($p < 0.05$) following the second dose of the Moderna vaccine.

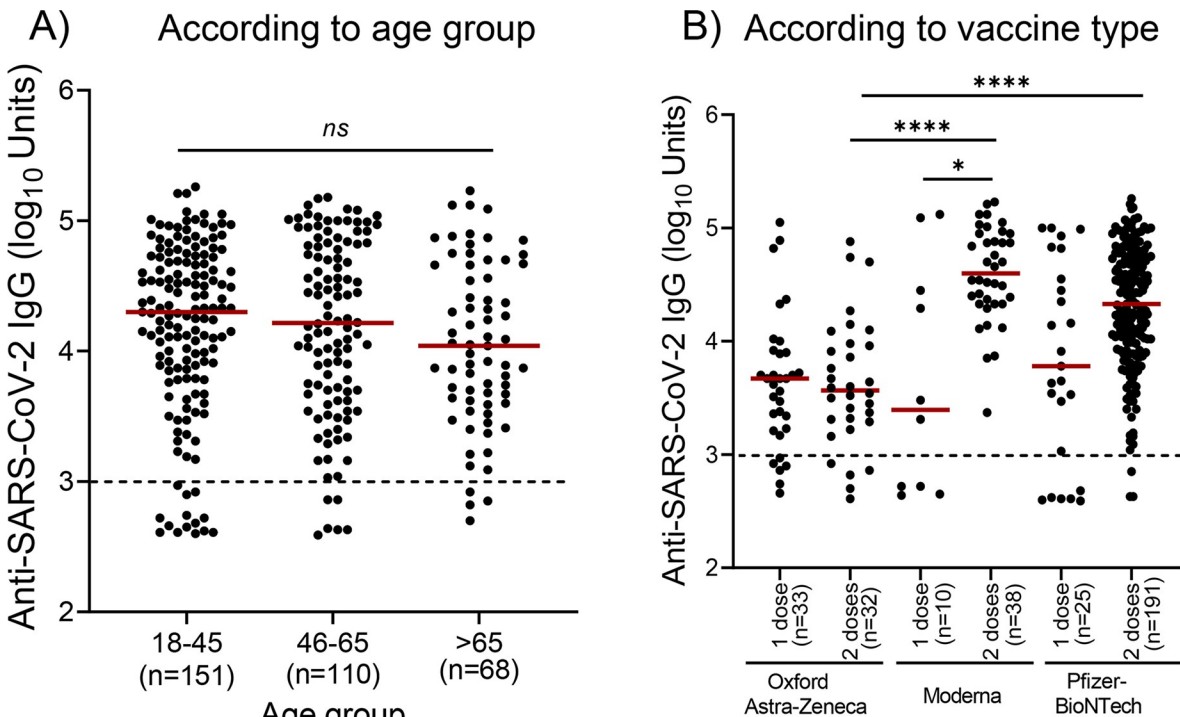

**Fig 2. Levels of IgG against SARS-CoV-2 in individuals without evidence of prior infection.** Participants received either one or two doses of a licensed vaccine. **A**) There were no significant differences in IgG in the three age groups. **B**) IgG levels according to vaccine type and the number of doses. Kruskal Wallis test followed by Dunn's multiple comparisons test was used for statistical analysis. Statistically significant differences are indicated with asterisks: $^*p<0.05$, $^{****}p<0.0001$, *ns*: non-significant. Horizontal bars indicate median values. Dotted lines: positive cut-off values.

Administration of a second dose of the Pfizer/BioNTech vaccine also resulted in the production of higher levels of IgG antibodies compared to the first dose, but these differences did not reach statistical significance ($p>0.05$). We then compared the levels of vaccine-elicited anti-spike IgG between the two different vaccine technologies, i.e., after the administration of i) one dose and ii) two doses of either an adenoviral-based vaccine (Oxford-Astra Zeneca) or an mRNA-based vaccine (Moderna or Pfizer-BioNTech). There was no statistically significant difference ($p>0.05$) in IgG levels between the groups who received a single dose of i) the Oxford- Astra Zeneca (n = 33) or Moderna (n = 10) vaccine and ii) the Oxford- Astra Zeneca (n = 33) and the Pfizer-BioNTech (n = 25) vaccine. There was also no statistically significant difference ($p>0.05$) in IgG levels between the groups who received a single dose of the Pfizer-BioNTech or Moderna vaccine. Anti-Spike IgG levels were significantly higher ($p<0.0001$) after two doses of Moderna (n = 38) than after Astra-Zeneca (n = 32). Vaccination with two doses of Pfizer/BioNTech (n = 191) induced the production of significantly higher levels ($p<0.0001$) of IgG antibodies than those produced after administration of two doses of Astra-Zeneca. However, there was no statistically significant difference ($p>0.05$) in the levels of anti-Spike IgG antibodies between the groups vaccinated with two doses of the Moderna or two doses of Pfizer-BioNTech (Fig 2B).

## Correlation of antibody levels post-vaccination with the age of the participants

Participants without evidence of prior infection who had received at least one dose of a vaccine were included in a subsequent analysis, to investigate associations between IgG levels and the

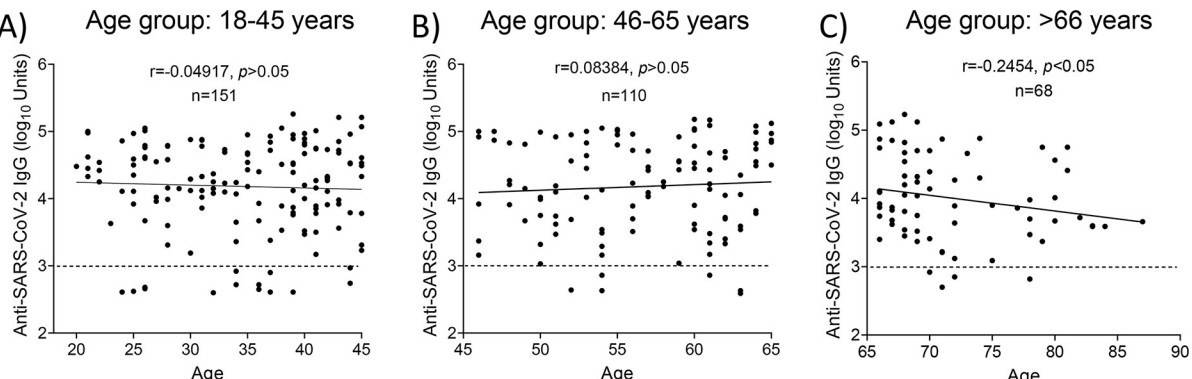

**Fig 3. Correlation of anti-SARS-CoV-2 IgG levels and age of participants without evidence of prior infection who had received at least one dose of a vaccine.** Correlation between IgG levels and the age of participants in three age groups namely 18–45 years (**A**), 46–65 years (**B**), and > 66years (**C**) was performed with Spearman's *r*-test. Dotted lines: positive cut-off values.

age of participants regardless of the type/doses of vaccine given. The differences in vaccine response by age group are presented in Fig 3. Spearman's *r*-test revealed that there was no correlation ($p>0.05$) between the anti-spike IgG levels and the age of participants in the 18–45 years (Fig 3A) and 46–65 years groups (Fig 3B), however, the levels of IgG in the >66 years of age group were negatively and weakly correlated ($r = -0.2454$; $p<0.05$) with the age of participants (Fig 3C).

## Comparison between the levels of SARS-CoV-2 antibodies elicited by SARS-CoV-2 vaccination and infection

As aforementioned, an important question is whether vaccine-induced responses are more potent and durable than those measured following natural infection. However, in the case of SARS-CoV-2, the extent to which infection can protect against subsequent reinfection remains unclear. Thus, we then sought to compare the levels of specific anti-SARs-CoV-2 IgG between vaccinated individuals without evidence of a previous infection with unvaccinated individuals with evidence of previous infection in the three age groups (Fig 4). Our analysis revealed that the levels of anti-spike IgG antibodies in vaccinated participants without evidence of prior infection were significantly higher compared to unvaccinated participants with evidence of prior infection in the 18–45 ($p<0.0001$) and 46–65 ($p<0.01$) age groups. However, there was no statistically significant difference ($p>0.05$) between the vaccine-elicited and infection-elicited antibodies in the >66 years of age group. Moreover, antibody responses in younger unvaccinated participants (18–45 years of age) with evidence of prior infection were lower compared to older participants (46–65 and >66 years of age group), but these differences did not reach statistical significance ($p>0.05$).

Moreover, in those without evidence of prior infection who received at least one dose of a vaccine, there was no statistically significant difference in anti-spike IgG antibodies levels between men and women in all age groups as resulted by the Mann Whitney *U*-test (S2 Fig). Finally, in unvaccinated individuals with evidence of prior infection, there was no statistically significant difference in IgG levels between men and women in all age groups as resulted by the Mann Whitney *U*-test (S3 Fig).

## Time course of anti-spike IgG antibodies after vaccination with one dose

The time courses of the anti-spike IgG levels in individuals without evidence of previous infection who received one dose of a licensed vaccine are illustrated in Fig 5(A)–5(C). An increase

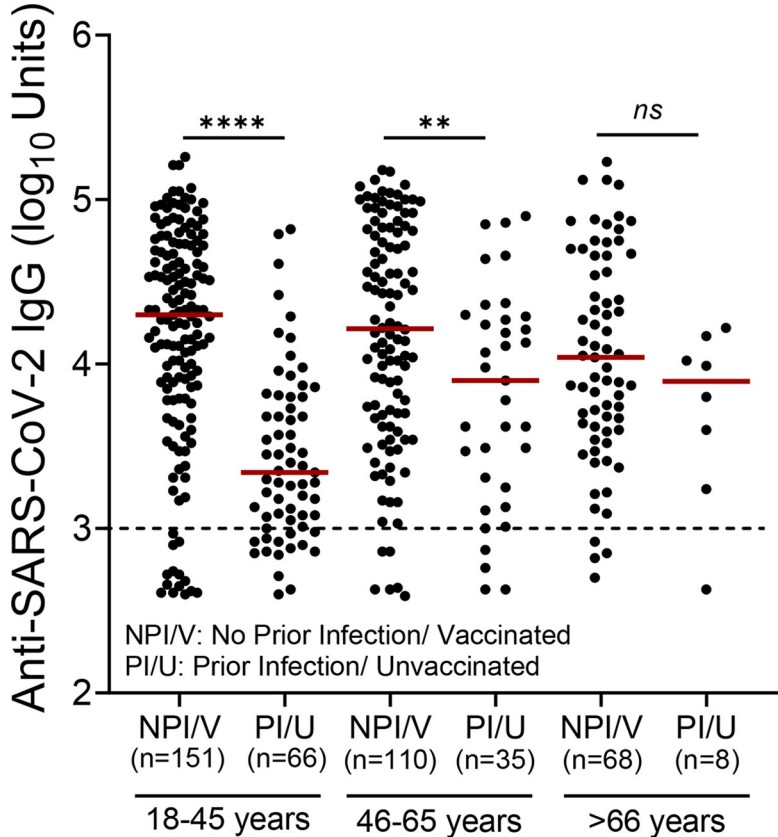

**Fig 4. Levels of anti-spike IgG across study groups.** Dot plots show the levels of anti-spike IgG in individuals without evidence of prior infection (NPI/V) who received one dose or two doses of a vaccine and unvaccinated individuals with evidence of prior infection (PI/U). The Mann-Whitney *U*-test was employed to compare the IgG levels in age groups. Statistically significant differences are indicated with asterisks: $^{**}p{<}0.01$, $^{****}p{<}0.0001$, *ns*: non-significant. Horizontal bars indicate median values.

in IgG levels was observed approximately 10 days post-vaccination regardless of the vaccine type (Fig 5A). Vaccinated individuals who had received one dose of an mRNA-based vaccine (Moderna or Pfizer/BioNTech) developed antibodies 10 days post-vaccination (Fig 5B), while high IgG titers were detected in individuals who received one dose of the Astra-Zeneca Vaccine (Fig 5C).

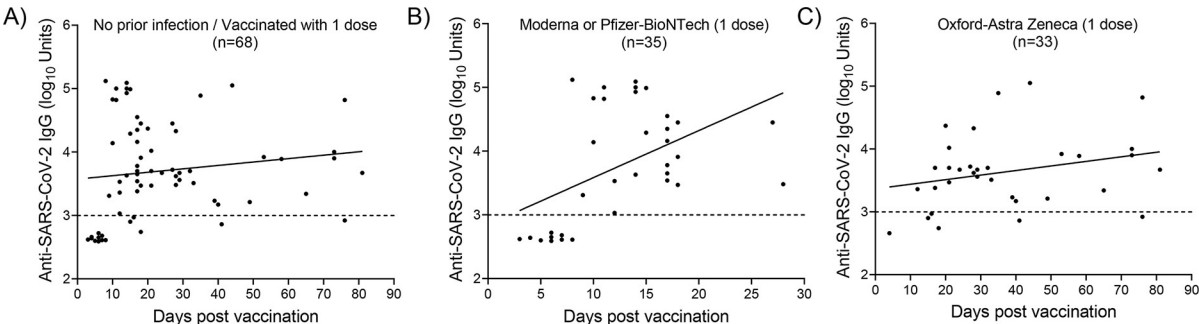

**Fig 5. Anti-spike IgG levels by time in individuals who received a single dose of a licensed vaccine. A)** Regardless of the vaccine type. **B)** mRNA-based vaccine (Moderna or Pfizer/BioNTech). **C)** Adenoviral-based vaccine (Oxford-Astra Zeneca).

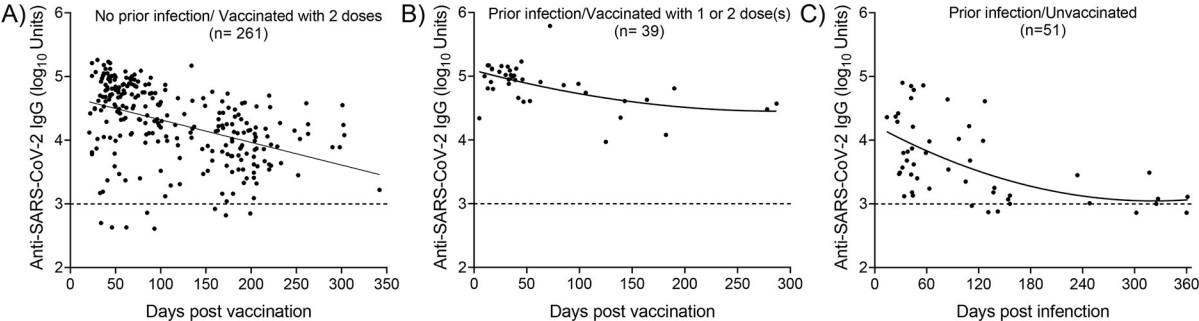

**Fig 6. Anti-spike IgG levels by time in different groups. A)** No prior infection and received two doses of a licensed vaccine. **B)** Prior infection and received one dose or two doses of a licensed vaccine. **C)** Prior infection without vaccination. For those who received two doses of a vaccine, the plots depict IgG levels from the date of the first vaccination.

## Time course of anti-spike IgG antibodies after vaccination and natural infection

The time courses of the anti-spike IgG levels (counting begins on the day of the first dose) in individuals without evidence of previous infection, who received two doses of a licensed vaccine (regardless of the vaccine type), as well as vaccinated and unvaccinated individuals with evidence of prior infection, are illustrated in Fig 6.

Anti-spike IgG antibodies were detectable in 96% of participants with no prior infection who had received two doses of a vaccine, while antibody levels showed a slight decrease 120 days post-vaccination (Fig 6A). Interestingly, antibody levels persisted in individuals with prior evidence of infection who received one dose or two doses of a vaccine (Fig 6B). Finally, in those with evidence of prior infection who have not been vaccinated, IgG levels showed a decrease approximately 120 days post-infection reaching the background levels 240 post-infection (Fig 6C). Taken together, these results revealed that vaccination induces antibody levels significantly higher and likely more durable compared to those produced after infection with SARS-CoV-2.

## Discussion

In this study, we assessed the immune response against SARS-CoV-2 in the general population in Cyprus between May and November 2021 by measuring the levels of specific anti-spike IgG antibodies and quantifying their levels over time. To the best of our knowledge, this is the first SARS-CoV-2 seroprevalence study carried out in Cyprus. The only other studies conducted in the Cypriot population examined the percentage of infected individuals in Cyprus who were able to produce antibodies against SARS-CoV-2 as well as the progression of SARS-CoV-2 antibody levels in SARS-CoV-2-infected individuals across time as a means of monitoring their antibody-mediated immunity after SARS-CoV-2 natural infection [25, 26].

A major question is whether there is a difference in the immunity conferred by natural SARS-CoV-2 infection vs. vaccination. Natural immunity is a feature of many viral infections, including, mumps, measles, and chickenpox, which induce remarkably stable neutralizing antibody responses [27]. The strength and duration of immunity after infection are key issues for 'shield immunity' [28], and for making informed decisions on how and when to ease physical distancing restrictions and other non-pharmacological interventions [29]. Previous studies have shown that circulating antibodies against SARS-CoV or MERS-CoV last for at least 1 year [30, 31]. Sustained IgG levels were maintained for more than 2 years after SARS-CoV infection [32, 33]. Antibody responses in individuals with laboratory-confirmed MERS-CoV

infection lasted for at least 34 months after the outbreak [34]. However, in the case of SARS-CoV-2, the extent to which infection can protect against subsequent reinfection is unclear [20]. In addition, if there is indeed protection against reinfection, how long-lasting it is, is currently a topic of intense discussion [35]. Nevertheless, recent studies demonstrate that SARS-CoV-2 mRNA vaccines elicit long-lived antibody responses that partially recognize and protect against antigenically distinct SARS-CoV-2 variants [36]. Consistent with several previous studies [37, 38], our analysis revealed that vaccines elicit higher levels of antibodies against SARS-CoV-2 compared to those elicited after infection with the virus (Figs 1 and 4). For the duration of the current study, the prevalent variants circulating in Cyprus were the Alpha (B.1.1.7), Delta (B.1.617.2), and Delta plus (AY.4.2) [39]. Therefore, data obtained most probably involve this particular variant and cannot be generalized for other SARS-CoV-2 variants such as the more recent Omicron (B.1.1.529).

Among individuals without evidence of previous infection, all age groups achieved high antibody levels after vaccination with one or two doses of a vaccine (Fig 2A), while two doses of an mRNA-based vaccine induced higher antibody levels compared to those induced after the administration of single-dose (Fig 2B). Furthermore, antibody responses in younger participants (18–45 years of age) without evidence of prior infection who received at least one dose of a licensed vaccine were higher compared to older participants, however, these differences did not reach statistical significance ($p > 0.05$) (Fig 2A).

Another important question is what happens when previously infected individuals are vaccinated. In this study, the highest levels of IgG were detected in the individuals with evidence of prior infection who had received one or two-dose of a vaccine. Recent studies by Stamatatos *et al*. [40] and Reynolds *et al*., [41] show that an impressive synergy occurs resulting from a combination of natural immunity and vaccine-generated immunity which is called "hybrid vigor immunity". When natural immunity to SARS-CoV-2 is combined with vaccine-generated immunity, a larger-than-expected immune response arises and our findings seem to lend further support to this.

In agreement with previous studies, IgG levels were decreased with older age (Fig 3). In a recent study, Wei et al [42] reported a non-linear correlation of anti-spike IgG positivity with age, while seropositivity dropped faster in participants of >75 years of age. In addition, recent studies revealed that antibody responses to SARS-CoV-2 could be detected in most infected individuals approximately 2 weeks after the onset of COVID-19 symptoms. However, due to the recent emergence of SARS-CoV-2 in the human population, it is not known how long antibody responses will be maintained or whether they will provide protection from reinfection [23]. Our analysis revealed that in the group without evidence of prior infection who received two doses of a vaccine, IgG levels showed a decrease 120–150 days post-vaccination (Fig 6A). A notable finding of this study was that antibody levels persisted in individuals with prior evidence of infection who received one dose or two doses of a vaccine (Fig 6B). However, in the unvaccinated group with evidence of prior infection, anti-IgG levels showed a sharp decrease approximately 100–120 days post-infection reaching the background levels after 240 days after infection (Fig 6C). The humoral response a few weeks after SARS-CoV-2 infection has been thoroughly described; some studies reported stable antibody levels within the first three months of recovery, whereas others showed a rapid decrease in convalescent patients [43]. Overall, our results suggest that two doses of vaccine lead to the induction of higher levels of long-lasting anti-spike IgG antibodies, while hybrid immunity to SARS-CoV-2 seems to be extremely potent. Recently Trougakos et al [44] demonstrated that natural infection promotes an earlier and more intense immune response compared to vaccination. However, the administration of a single dose of a vaccine to individuals previously infected with SARS-CoV-2

(hybrid immunity), or two doses in non-infected individuals, induces antibody levels significantly higher and likely more durable compared to natural infection.

Our study has some limitations. The limited information provided by participants of the study does not allow us to distinguish between differences in disease severity in the category of previously infected individuals. This category includes participants that could have been asymptomatic with just a positive molecular test, to have mild or severe disease. Additionally, this study was designed to only monitor for IgG antibodies and no other types such as IgA, etc. Additionally, utilizing more than one test/kit in future experiments can aid in determining the validity of the status of the samples for SARS-CoV-2 antibodies. It should be pointed out that determining antibody levels against SARS-CoV-2 could be useful in epidemiological studies, for estimating the spread of the infection and the lethality rate, in the serological diagnosis of individuals with mild or moderate symptoms, and those who are asymptomatic, in the first screening of convalescent patients for plasma collection and the monitoring of the antibody response of vaccinated subjects [36].

One strength of our study is that our tested population was drawn by random sampling, therefore minimizing the sampling bias on the estimation of seroprevalence in our population. Furthermore, measuring serum/plasma levels of virus-specific antibodies to SARS-CoV-2 could be used as an alternative method for detecting COVID-19 infection or as complementary tests, in addition to RT-PCR. Compared with the sampling methods required for RT-PCR, this serological assay reduces the risk of aerosol exposure, making it safer for medical staff to use. However, a limitation of antibody tests is that they require a longer window period after infection than RT-PCR [45]. Serological testing may be helpful for the diagnosis of suspected patients with negative RT-PCR results and the identification of asymptomatic infections [43].

## Conclusions

Predicting the long-term potential for immune control of SARS-CoV-2 is challenging. Our results and those of others [40, 41] suggest that hybrid immunity to SARS-CoV-2 appears to be impressively potent. Our results also suggest that vaccine-induced responses are significantly more effective than natural immunity alone. Interestingly, and in consistence with previous studies, our results indicate that the vaccination of previously infected individuals drives a rapid and very potent recall of humoral immunity, even after a single vaccine dose.

## Supporting information

**S1 Fig. Comparison of anti-spike IgG antibody responses between men and women in the groups without or with evidence of prior infection.** Participants were not vaccinated (UnVac) or received at least one dose of a vaccine against SARS-CoV-2 (Vac). The number of participants (n) in each group is shown in parentheses. Mann-Whitney *U*-test was used for pairwise statistical analysis. There were no significant (*ns*) differences in anti-spike IgG antibody responses between men and women in all groups. Horizontal bars indicate median values. Dotted line: positive cut-off value.
(TIF)

**S2 Fig. Comparison of anti-spike IgG antibody responses between men and women in the age groups 18–45 years, 46–65 years, and >65 years following vaccination with at least one dose of a vaccine against SARS-CoV-2.** The number of participants (n) in each group is shown in parentheses. Mann-Whitney *U*-test was used for pairwise statistical analysis. There were no significant (*ns*) differences in anti-spike IgG antibody responses between men and

women in all groups. Horizontal bars indicate median values. Dotted line: positive cut-off value.
(TIF)

**S3 Fig. Comparison of anti-spike IgG antibody responses between men and women in the age groups 18–45 years, 46–65 years, and >65 years after infection with SARS-CoV-2.** All participants were not vaccinated against SARS-CoV-2. The number of participants (n) in each group is shown in parentheses. Mann-Whitney *U*-test was used for pairwise statistical analysis. There were no significant (*ns*) differences in anti-spike IgG antibody responses between men and women in all groups. Horizontal bars indicate median values. Dotted line: positive cut-off value.
(TIF)

## Acknowledgments

We would like to express our special thanks to SIEMENS Healthcare which technical support throughout the study.

## Author Contributions

**Conceptualization:** Christos Papaneophytou, Tasos Kalogiannis, Kyriacos Yiannoukas, Christos C. Petrou.

**Data curation:** Christos Papaneophytou, Andria Nicolaou, Myrtani Pieri, Vicky Nicolaidou, Eleftheria Galatou, Yiannis Sarigiannis, Markella Pantelidou, Pavlos Panayi, Theklios Thoma, Antonia Stavraki, Tasos Kalogiannis, Kyriacos Yiannoukas, Christos C. Petrou, Kyriacos Felekkis.

**Formal analysis:** Christos Papaneophytou, Andria Nicolaou, Myrtani Pieri, Vicky Nicolaidou, Yiannis Sarigiannis, Pavlos Panayi, Antonia Stavraki, Xenia Argyrou, Tasos Kalogiannis, Kyriacos Yiannoukas, Christos C. Petrou, Kyriacos Felekkis.

**Funding acquisition:** Tasos Kalogiannis, Kyriacos Yiannoukas, Christos C. Petrou, Kyriacos Felekkis.

**Investigation:** Andria Nicolaou, Myrtani Pieri, Vicky Nicolaidou, Eleftheria Galatou, Yiannis Sarigiannis, Markella Pantelidou, Theklios Thoma, Antonia Stavraki, Xenia Argyrou, Kyriacos Yiannoukas, Christos C. Petrou, Kyriacos Felekkis.

**Methodology:** Christos Papaneophytou, Andria Nicolaou, Myrtani Pieri, Vicky Nicolaidou, Yiannis Sarigiannis, Markella Pantelidou, Pavlos Panayi, Theklios Thoma, Kyriacos Felekkis.

**Project administration:** Christos Papaneophytou, Tasos Kalogiannis, Kyriacos Yiannoukas, Christos C. Petrou, Kyriacos Felekkis.

**Resources:** Kyriacos Felekkis.

**Supervision:** Kyriacos Felekkis.

**Validation:** Christos Papaneophytou, Myrtani Pieri, Vicky Nicolaidou, Yiannis Sarigiannis, Tasos Kalogiannis, Kyriacos Yiannoukas, Christos C. Petrou, Kyriacos Felekkis.

**Visualization:** Kyriacos Felekkis.

**Writing – original draft:** Christos Papaneophytou, Kyriacos Felekkis.

**Writing – review & editing:** Christos Papaneophytou, Kyriacos Felekkis.

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
