## [Decision Letter · Decision Letter 0]

27 Apr 2022

PONE-D-22-04892Seroprevalence of immunoglobulin G antibodies against SARS-CoV-2 in CyprusPLOS ONE

Dear Dr. Felekkis,

Thank you for submitting your manuscript to PLOS ONE. While both reviewers consider the results of this study valuable and interesting for the field, they have raised a number of concerns (see the comments listed below).  Therefore, we invite you to submit a revised version of the manuscript that addresses the points raised during the review process, in particular, the concern on experimental details of IgG titration and antigen used for detection of IgG should be addressed.

We look forward to receiving your revised manuscript.

Kind regards,

Tian Wang, PhD

Academic Editor

PLOS ONE

Journal Requirements:

Reviewers' comments:

Reviewer's Responses to Questions

**Comments to the Author**

1. Is the manuscript technically sound, and do the data support the conclusions?

Reviewer #1: Yes

Reviewer #2: Yes

2. Has the statistical analysis been performed appropriately and rigorously? 

Reviewer #1: N/A

Reviewer #2: Yes

3. Have the authors made all data underlying the findings in their manuscript fully available?

Reviewer #1: Yes

Reviewer #2: Yes

4. Is the manuscript presented in an intelligible fashion and written in standard English?

Reviewer #1: Yes

Reviewer #2: Yes

5. Review Comments to the Author

Reviewer #1: Comments to the author:

In this manuscript, the authors used blood samples from 702 participants from four cities in Cyprus (Europe) to estimate the seroprevalence of SARS-CoV2 IgG antibody and to compare the antibody levels among three groups of participants including: i) vaccinated without evidence of the previous infection, ii) unvaccinated with evidence of the previous infection, and iii) vaccinated with evidence of the previous infection. The results from this study revealed that the highest response of virus-specific IgG antibodies were observed in individuals who were infected with SARS-CoV-2 and received at least one dose of a vaccine. The results also indicate that vaccine-induced responses lead to higher IgG antibodies response compared to those produced following infection with the virus.

I find the study noteworthy because 1) the data overall for SARS-CoV2 infection in Cyprus is limited, and 2) the study employed a large sample size and different groups which strengthens the study. Here are several points that need to be considered:

Major points:

- I would suggest analyzing some of the SARS-CoV2 IgG positive samples in the viral neutralization assay to determine the magnitude of neutralizing antibodies (NAbs).

- The manuscript lack samples from different time points. Therefore, I would suggest the authors include flow-up samples if that is possible for the durability of the IgG responses.

Minor points:

- The abstract section is poorly written. Therefore I would suggest the authors improve the abstract by summarizing/revising the current abstract.

- What are the methods used to determine the previous SARS-2. I encourage the author to state that in the methods section.

- Since the blood samples were collected in the anticoagulant tube, samples should be plasma not serum samples, I encourage the authors correct that in the method section.

- P6 and P7: in the methods section, subtitle laboratory testing, I suggest the authors to describe the assays in this section separately, for instance, ELISA should be in the separate subtitle with more details..……..etc.

- The specificity and sensitivity of the ELISA assay should be mentioned in the method section.

- Since the collected samples from participants visited the clinic for routine check-up or other tests, Are the patients have SARS-CoV2 acute symptoms? If so, are the samples tested for the IgM antibody? It would be nice if the authors include the IgM data since some samples maybe collected during the acute phase of infection.

- The authors should describe how they determine the antibody level somewhere in the manuscript. otherwise, the term lgG level should be changed to the IgG titer if they already checked the titer if not they should write IgG response.

- The IgG subclass need to be tested at least for the samples that show high IgG response.

- The study inclusion/exclusion criteria in general need further explanation for the reader. For example, were subjects with COVID-19 symptoms excluded? What tests were performed to exclude other common causes of COVID-19 disease-like symptoms?

- P20L456-457: The authors stated that their results also “suggest that vaccine-induced responses are significantly more effective than natural immunity alone” I think this statement needs to be discussed /explained in more detail in the discussion section.

- The discussion should describe the findings in a more concise manner.

In general, this work deals with an important topic and provides useful information to the reader. However, given all the above concerns, I believe this manuscript requires fixing the points before being accepted for publication.

Reviewer #2: In this study, the authors performed sera analysis of SARS-COV2 -IgG levels among unvaccinated, infected and vaccinated populations in Cyprus. Aging, vaccine type, Vaccine time course and natural infection have been used as parameters of comparison. The main conclusion of this study is that vaccine-induced higher IgG responses compared to SARS-COV2 infection. Overall, the manuscript is nicely written and the conclusions were supported by the results. The results are interesting to the field. There are a few concerns to be addressed:

1) Methods section: Information for the SARS-COV2 spike protein used for the detection of IgG responses is not clearly described, such as resource of the antigen, etc.

2) Methods section: it’s not clear how the IgG titers were determined. What’s the cutoff?

3) The abstract section needs to be more concise.

4) Line 354: please add “days” after “120”

6. PLOS authors have the option to publish the peer review history of their article (what does this mean?). If published, this will include your full peer review and any attached files.

Reviewer #1: 

Reviewer #2: No

---

## [Author Response · Author response to Decision Letter 0]

4 May 2022

Our revised manuscript meets PLOS ONE's style requirements including those for file naming. 

The study was supported internally by the University of Nicosia and Yiannoukas Medical Laboratories/ Bioiatriki Group 

b) State what role the funders took in the study. If the funders had no role in your study, please state: 

The authors did not receive any salary from the funders as part of this work.

We have already provided the DOI to access our data (P.21, LL.481-482):

“Dataset are deposited in Zenodo and can be accessed through the following link: https://doi.org/10.5281/zenodo.5992410”

We remove the phrase “data not shown” and we are providing these results as supplementary figures (S1-S3). 

The ethics statement appears only in the Methods section of the revised manuscript (P.6, LL. 136-137)

Reviewers' comments

Reviewer #1

In this manuscript, the authors used blood samples from 702 participants from four cities in Cyprus (Europe) to estimate the seroprevalence of SARS-CoV2 IgG antibody and to compare the antibody levels among three groups of participants including: i) vaccinated without evidence of the previous infection, ii) unvaccinated with evidence of the previous infection, and iii) vaccinated with evidence of the previous infection. The results from this study revealed that the highest response of virus-specific IgG antibodies were observed in individuals who were infected with SARS-CoV-2 and received at least one dose of a vaccine. The results also indicate that vaccine-induced responses lead to higher IgG antibodies response compared to those produced following infection with the virus. 

I find the study noteworthy because 1) the data overall for SARS-CoV2 infection in Cyprus is limited, and 2) the study employed a large sample size and different groups which strengthens the study. Here are several points that need to be considered: 

Major points: 

1. I would suggest analyzing some of the SARS-2 IgG positive samples in the viral neutralization assay to determine the magnitude of neutralizing antibodies (NAbs). 

This is an interesting and valid pont. However, due to budget limitations we were not able to perform experiments to determine the magnitude of neutralizing antibodies (NAbs) in the recruited subjects. 

2. The manuscript lacks samples from different time points. Therefore, I would suggest the authors include flow-up samples if that is possible for the durability of the IgG responses. 

Indeed, it would be interesting to assess the durability of the IgG responses. However, this study did not include follow-up experiments. The approval we obtained from the Cyprus Bioethics Committee allowed us to collect samples from participants only one time. 

Minor points: 

1. The abstract section is poorly written. Therefore, I would suggest the authors improve the abstract by summarizing/revising the current abstract. 

We thank the Reviewer for this valuable comment. We revised the abstract and reduced its length. 

2. What are the methods used to determine the previous SARS-2. I encourage the author to state that in the methods section. 

We thank the reviewer for this comment. The methods used (i.e., antigen rapid test or RT-qPCR test) are mentioned in the revised manuscript. P.6, LL.145-146. 

3. Since the blood samples were collected in the anticoagulant tube, samples should be plasma not serum samples, I encourage the authors correct that in the method section. 

The Reviewer is right. We replaced “serum” with “plasma” in the revised manuscript.

4. P6 and P7: in the methods section, subtitle laboratory testing, I suggest the authors to describe the assays in this section separately, for instance, ELISA should be in the separate subtitle with more details...…….etc. 

This section has been corrected according to the Reviewer’s instructions. 

5. The specificity and sensitivity of the ELISA assay should be mentioned in the method section. 

The specificity and sensitivity of the ELISA assay are mentioned in the revised manuscript (P.8, LL.180-182). 

“The sensitivity of the method (for samples obtained ≥ 14 days post-infection with SARS-CoV-2) was 100% (95% CI, 95.9-100%) and the specificity was 100% (95% CI, 95.8-100%).”

6. Since the collected samples from participants visited the clinic for routine check-up or other tests, Are the patients have SARS-CoV2 acute symptoms? If so, are the samples tested for the IgM antibody? It would be nice if the authors include the IgM data since some samples maybe collected during the acute phase of infection. 

We agree with the Reviewer. However, in this study subjects with COVID-19 symptoms were excluded from this study and only individuals with a negative antigen rapid test and/or RT-qPCR test were eligible to participate in this study.

7. The authors should describe how they determine the antibody level somewhere in the manuscript. otherwise, the term lgG level should be changed to the IgG titer if they already checked the titer if not, they should write IgG response. 

We thank the Reviewer for this comment. We added the following paragraph (PP.7-8, LL.175-180) in the revised manuscript to explain how we determined the antibody levels (in relative units):

“The resulting signal was measured at 612nm and was proportional to IgG concertation in the sample. IgG levels were determined by the semiquantitative mode of the SIEMENS Dimension EXL system using a 5-level LOGIT calibration curve and the results were presented as relative (Ind) Units. The Dimension EXL system cutoff analyte value was 1000 relative Units and it was used to identify IgG positive samples”. 

8. The IgG subclass need to be tested at least for the samples that show high IgG response. 

We thank the Reviewer for this interesting comment. Indeed, it would be interesting to test the IgG subclass in the samples containing high IgG levels. However, we were not able to perform these experiments due to budget limitations. 

9. The study inclusion/exclusion criteria in general need further explanation for the reader. For example, were subjects with COVID-19 symptoms excluded? What tests were performed to exclude other common causes of COVID-19 disease-like symptoms? 

Subjects with COVID-19 symptoms were excluded from this study and only individuals with a negative antigen rapid test and/or RT-qPCR test were eligible to participate in this study. We also mentioned this limitation in the revised manuscript (P.6, LL.146-148). 

“Furthermore, only participants with a negative antigen rapid test and/or an RT-qPCR test for SARS-CoV-2 at the time of blood collection were eligible to enroll in this study.

10. P20, L456-457: The authors stated that their results also “suggest that vaccine-induced responses are significantly more effective than natural immunity alone” I think this statement needs to be discussed /explained in more detail in the discussion section. 

This statement is extensively discussed in the discussion section (P18. LL.398-405) as follows:

 “ However, in the case of SARS-CoV-2, the extent to which infection can protect against subsequent reinfection is unclear (20). In addition, if there is indeed protection against reinfection, how long-lasting it is, is currently a topic of intense discussion(35). Nevertheless, recent studies demonstrate that SARS-CoV-2 mRNA vaccines elicit long-lived antibody responses that partially recognize and protect against antigenically distinct SARS-CoV-2 variants (36). Consistent with several previous studies (37, 38), our analysis revealed that vaccines elicit higher levels of antibodies against SARS-CoV-2 compared to those elicited after infection with the virus (Fig 1 and 4).”

11. The discussion should describe the findings in a more concise manner. 

We appreciate Reviewer’s comment. We feel that the above changes will make the description of the findings more concice. We strongly believe that if we further modify the content of the discussion session it would weaken our findings. 

In general, this work deals with an important topic and provides useful information to the reader. However, given all the above concerns, I believe this manuscript requires fixing the points before being accepted for publication.

 Thank you for your valuable comments and effort.

Reviewer #2: 

In this study, the authors performed sera analysis of SARS-COV2 -IgG levels among unvaccinated, infected and vaccinated populations in Cyprus. Aging, vaccine type, Vaccine time course and natural infection have been used as parameters of comparison. The main conclusion of this study is that vaccine-induced higher IgG responses compared to SARS-COV2 infection. Overall, the manuscript is nicely written and the conclusions were supported by the results. The results are interesting to the field. There are a few concerns to be addressed:

1. Methods section: Information for the SARS-COV2 spike protein used for the detection of IgG responses is not clearly described, such as resource of the antigen, etc.

More information about the spike protein used for the detection of IgG responses has been added in the revised manuscript. (P.7, LL.165-170.)

“The LOCI reagents include two synthetic bead reagents (Sensibeads and Chemibeads) and a biotinylated mouse anti-human IgG antibody. Sensibeads are coated with streptavidin and contain photosensitizer. Chemibeads are coated with anti-‑ Fluorescein isothiocyanate (FITC) antibody and contain chemiluminescent dye. Furthermore, the anti-FITC antibody-coated-Chemibeads are pre-decorated with fluoresceinated S1 receptor-binding domain (RBD) antigen of the spike protein of SARS-CoV-2 virus.”

2. Methods section: it’s not clear how the IgG titers were determined. What’s the cutoff?

We thank the Reviewer for this comment. We added the following paragraph (PP.8, LL.175-170) in the revised manuscript to explain how we determined the antibody levels (in relative units):

“The resulting signal was measured at 612nm and was proportional to IgG concertation in the sample. IgG levels were determined by the semiquantitative mode of the SIEMENS Dimension EXL system using a 5-level LOGIT calibration curve and the results were presented as relative (Ind) Units. The Dimension EXL system cutoff analyte value was 1000 relative Units and it was used to identify IgG positive samples”. 

3. The abstract section needs to be more concise.

We thank the Reviewer for this valuable comment. We revised the abstract and reduced its length. 

4. Line 354: please add “days” after “120”

Corrected

---

## [Decision Letter · Decision Letter 1]

17 May 2022

PONE-D-22-04892R1Seroprevalence of immunoglobulin G antibodies against SARS-CoV-2 in CyprusPLOS ONE

Dear Dr. Felekkis,

Thank you for submitting your revised manuscript to PLOS ONE. Although both reviewers felt the manuscript has been improved significantly, there are some remaining concerns to be addressed (see the comments below). Therefore, we invite you to submit a revised version of the manuscript that addresses the points raised during the review process.

We look forward to receiving your revised manuscript.

Kind regards,

Tian Wang, PhD

Academic Editor

PLOS ONE

Journal Requirements:

Reviewers' comments:

Reviewer's Responses to Questions

**Comments to the Author**

1. If the authors have adequately addressed your comments raised in a previous round of review and you feel that this manuscript is now acceptable for publication, you may indicate that here to bypass the “Comments to the Author” section, enter your conflict of interest statement in the “Confidential to Editor” section, and submit your "Accept" recommendation.

Reviewer #1: All comments have been addressed

Reviewer #2: (No Response)

2. Is the manuscript technically sound, and do the data support the conclusions?

Reviewer #1: Yes

Reviewer #2: Yes

3. Has the statistical analysis been performed appropriately and rigorously? 

Reviewer #1: Yes

Reviewer #2: Yes

4. Have the authors made all data underlying the findings in their manuscript fully available?

Reviewer #1: Yes

Reviewer #2: Yes

5. Is the manuscript presented in an intelligible fashion and written in standard English?

Reviewer #1: (No Response)

Reviewer #2: Yes

6. Review Comments to the Author

Reviewer #1: Comments to the author:

This work deals with an important topic and provides useful information to the reader. I believe the manuscript has been improved a lot.

Minor points:

- Some areas of the manuscript require significant proofreading and revision is needed to improve the quality of English for instance the abstract section has unclear sentences.

- Source of the reagents should be provided in the method section (antibody measurement section).

Reviewer #2: The authors have partially addressed my concerns. Here are the remaining concerns to be addressed:

1. Line 170: Please include the resource (company) where the fluoresceinated S1 receptor-binding RBD antigen was purchased, including the catalog #. If the antigen was made in house, please include the sequence information.

2. Line 29: please add “,” after this study, and change “plasma the” to “the plasma”.

3. Line 42: please change “humoral immunity” to “antibody responses”

7. PLOS authors have the option to publish the peer review history of their article (what does this mean?). If published, this will include your full peer review and any attached files.

Reviewer #1: No

Reviewer #2: No

---

## [Author Response · Author response to Decision Letter 1]

22 May 2022

Reviewer #1: 

Comments to the author:

This work deals with an important topic and provides useful information to the reader. I believe the manuscript has been improved a lot.

Minor points:

- Some areas of the manuscript require significant proofreading and revision is needed to improve the quality of English for instance the abstract section has unclear sentences.

The manuscript was reviewed and revised accordingly.

- Source of the reagents should be provided in the method section (antibody measurement section).

The source of reagents is provided in the revised manuscript (Siemens Healthcare Diagnostics Inc., Erlangen, Germany).

Reviewer #2: 

The authors have partially addressed my concerns. Here are the remaining concerns to be addressed:

1. Line 170: Please include the resource (company) where the fluoresceinated S1 receptor-binding RBD antigen was purchased, including the catalog #. If the antigen was made in house, please include the sequence information. 

The following text was added to the manuscript to address reviewer’s comment

“All components for the detection of anti-IgG against SARS-CoV-2 including the fluoresceinated S1 receptor-binding RBD antigen were included in the Dimension Vista SARS CoV 2 IgG (COV2G) assay (Siemens Healthcare Diagnostics Inc., Erlangen, Germany, Cat #K7771

11417771)”

2. Line 29: please add “,” after this study, and change “plasma the” to “the plasma”.

Corrected 

3. Line 42: please change “humoral immunity” to “antibody responses”

Corrected

---

## [Editor Report · Decision Letter 2]

30 May 2022

Seroprevalence of immunoglobulin G antibodies against SARS-CoV-2 in Cyprus

PONE-D-22-04892R2

Dear Dr. Felekkis,

We’re pleased to inform you that your manuscript has been judged scientifically suitable for publication and will be formally accepted for publication once it meets all outstanding technical requirements.

Kind regards,

Tian Wang, PhD

Academic Editor

PLOS ONE
---

## [Editor Report · Acceptance letter]

3 Jun 2022

PONE-D-22-04892R2 

Seroprevalence of immunoglobulin G antibodies against SARS-CoV-2 in Cyprus 

Dear Dr. Felekkis:

I'm pleased to inform you that your manuscript has been deemed suitable for publication in PLOS ONE. Congratulations! Your manuscript is now with our production department. 

Kind regards, 

on behalf of

Dr. Tian Wang 

Academic Editor

PLOS ONE